# MANY CAN BEAT ONE: MoE-LINEAR ATTENTION FOR FULL MLP IMAGE GENERATION

## ABSTRACT

Although Transformer-based models have achieved significant success in image generation tasks, the computation of scaled dot-product attention for token interactions incurs substantial computational overhead. To address this issue, researchers have attempted to directly optimize the attention matrix using methods like gradient descent, treating the attention matrix as a set of learnable parameters. However, the attention matrix learned through this approach aims to capture a global interaction pattern. Specifically, for all input images, the tokens interact based on a single learned attention matrix. Since the distribution, size, and other characteristics of objects in each image can vary, the attention matrix learned in this way is often suboptimal. To overcome this limitation, we propose MoE-MLP, which introduces two novel components: **1) MoE-Linear Attention Module:** We design multiple learnable attention matrices and adaptively assign a weight to each matrix for every image. These matrices are then linearly combined to form the final attention matrix. Given that there are numerous possible combinations of weights, the model can learn a more suitable combination for each image; **2) Multi-Head Module:** We partition the original channels into several heads and perform MoE-Linear Attention on each head separately. This significantly increases the diversity of attention matrix combinations for different images. Finally, we conduct experiments on MS-COCO datasets, and the results demonstrate that our method achieves 7.43 FID (with **1.19** improvement), which significantly outperforms traditional MLP-based approaches (8.62 FID), with only negligible additional computational cost.

## 1 INTRODUCTION

Transformers (Vaswani et al., 2017) have dominated both natural language processing (Kenton & Toutanova, 2019; Touvron et al., 2023a;b), computer vision (Dosovitskiy et al., 2021; Pu et al., 2022; Li et al., 2022b), and multi-modal (Li et al., 2022a; 2023; Lin et al., 2024; Liu et al., 2023) tasks. These model architectures have demonstrated exceptional scalability and flexibility, replacing traditional models such as RNNs (Gregor et al., 2015) and CNNs (Odena et al., 2017; Goodfellow et al., 2014; Gulrajani et al., 2017). The success of the Transformer lies in its multi-head self-attention mechanism (Vaswani et al., 2017), which facilitates token-wise information interaction by using scaled dot-product attention. Despite the impressive expressive power of the Transformer, the self-attention mechanism is computationally expensive and complex. Furthermore, this mechanism has no direct counterpart in biological neural networks, as the human brain does not perform token-wise dot-product operations (Hu & Rostami, 2024).

To address the computational challenges, some works (Touvron et al., 2022; Tolstikhin et al., 2021; Hou et al., 2022; Yu et al., 2022) attempt to directly optimize the attention matrix using gradient descent, treating the attention matrix as a set of learnable parameters. While full MLP-based models perform well in simpler tasks, such as image classification (Tolstikhin et al., 2021), they struggle to achieve Transformer-comparable results in more complex tasks, such as image generation (Hu & Rostami, 2024). The reason lies in the fact that these models aim to learn a global token interaction pattern *i.e.*, an attention matrix that applies to all images within the distribution. However, due to the significant variations in object positions, sizes, and other features across images, it is difficult to apply a single, global token interaction pattern to all images, leading to suboptimal performance.

To address the limitations of the single token interactions pattern, we propose two novel modules: the MoE-Linear Attention module and the Multi-Head module. Specifically, for the **MoE-Linear**

**Attention module**, we design several learnable attention matrices and adaptively assign learnable weights to each matrix for every image. These matrices are then linearly combined to form the final attention matrix (different image has different combination weights). As illustrated in Figure 1, even with just two learnable attention matrices, the model can adaptively learn an infinite number of linear combinations. For the **Multi-Head module**, we partition the original channels into multiple heads and perform MoE-Linear Attention on each head separately (different heads do not share the learnable attention matrices), significantly increasing the diversity of attention matrix combinations across different images. The main contributions of this paper can be summarized as follows:

**1)** We introduce the MoE-MLP, which includes the MoE-Linear Attention module and the Multi-Head module, significantly enhancing the diversity of token interaction patterns in full MLP models.

**2)** By leveraging the property of linear combination, we reduce the complexity of MoE-Linear Attention from $EHL^2D$ to $L^2D + HELD$, where $E$, $H$, $L$, and $D$ are the number of experts, the number of heads, sequence length, and hidden dimension, respectively. Experimentally, our MoE-MLP (2 Experts 4 Heads) only brings about 0.01 extra GFLOPs for each MoE-MLP block.

**3)** We conduct extensive text-to-image experiments on the MS-COCO dataset, and our results demonstrate that our approach achieves new state-of-the-art performance within the MLP architecture.

## 2 RELATED WORKS

In this section, we briefly review works about visual MLP and transformer-based diffusion models.

**Visual MLP models.** MLP-based models (Tolstikhin et al., 2021; Yu et al., 2022) have gained attention as a strong alternative to traditional CNNs and ViTs for vision tasks in the past years. By leveraging the simplicity and computational efficiency of MLPs, these models achieve competitive results across a range of applications. The MLP-Mixer (Tolstikhin et al., 2021) was among the first to propose the idea of mixing tokens and channels via separate MLPs, challenging the conventional reliance on convolutions and attention mechanisms for high performance in visual classification. gMLP (Liu et al., 2021) enhanced the MLP framework by introducing gating mechanisms, which improved gradient propagation and increased model expressiveness. resMLP (Touvron et al., 2022) addressed the vanishing gradient issue by integrating residual connections into the MLP design, enabling the training of deeper networks. S2MLP (Yu et al., 2022) introduced a spatial-shift operation, which better captures spatial relationships between pixels. CycleMLP (Chen et al., 2022) utilized cyclic shifting to efficiently capture long-range dependencies and contextual information. Vision Permutator (ViP) (Hou et al., 2022) proposed a novel permutation-based approach that enables the model to permute the input image, allowing it to learn complex patterns in the data. These MLP-based models have primarily been applied to simple tasks *e.g.*, image classification, where the moderate information loss is acceptable. However, when applying them to more complex tasks, *e.g.*, image generation (Hu & Rostami, 2024), these models have not been shown comparable results to Transformers, and even worse than CNN (Rombach et al., 2022).

**Transformer-based diffusion architectures.** Recently, there has been a notable shift towards adopting Transformer-based (Chen et al., 2024; Peebles & Xie, 2023; Bao et al., 2023) architectures, which are gradually replacing the traditional UNet architecture (Ho et al., 2020; Ronneberger et al., 2015; Nichol & Dhariwal, 2021) in both image and video generation tasks (Yang et al., 2025). These Transformer-based models leverage powerful attention mechanisms to better capture long-range dependencies and hierarchical structures within the data, leading to improved performance and flexibility in generating diverse visual content. Among these works, U-ViT (Bao et al., 2023) and DiT (Peebles & Xie, 2023) propose the long skip connection and adaptive layer normalization to enhance the generated quality and controllability, respectively. The two methods provided a solid foundation for later transformer-based diffusion approaches (Esser et al., 2024; Crowson et al., 2024).

## 3 METHODS

### 3.1 PRELIMINARY

**Diffusion Models.** These models (e.g. the seminal work (Ho et al., 2020; Lu et al., 2022)) gradually inject noise into data and then reverse this process to generate data from noise. The noise-injection process is also called the forward process. Given clean data $\mathbf{x}_0$, the forward process can be written as:

$$q(\mathbf{x}_{1:T}|\mathbf{x}_0) = \prod_{t=1}^{T} q(\mathbf{x}_t|\mathbf{x}_{t-1}) \tag{1}$$

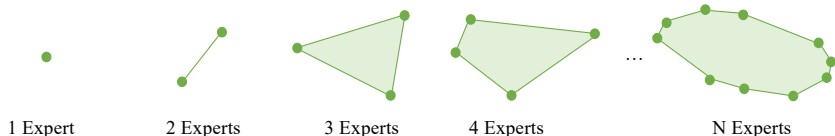

1 Expert     2 Experts     3 Experts     4 Experts     N Experts

Figure 1: Motivation of the MoE architecture. We observed that traditional Transformer-free approaches (Hu & Rostami, 2024; Tolstikhin et al., 2021) use a single matrix for information propagation between tokens (we represent each learnable attention matrix with a single dot). However, a single matrix struggles to adapt to different images (as different images contain varying content). In contrast, our proposed method leverages MoE to learn weighted combinations of different matrices, thereby enabling unlimited kinds of attention matrices to better adapt to diverse images.

where $q$ is the forward process and $q(\mathbf{x}_t|\mathbf{x}_{t-1}) = \mathcal{N}(\mathbf{x}_t|\sqrt{\alpha_t}\mathbf{x}_{t-1}, \beta_t\mathbf{I})$, and $\alpha$ and $\beta$ represent the noise schedule and $\alpha + \beta = 1$. $\mathcal{N}(0, 1)$ means the standard Gaussion noise. To reverse this process, a Gaussion model $p(\mathbf{x}_{t-1}|\mathbf{x}_t) = \mathcal{N}(x_{t-1}|\mu_t(\mathbf{x}_t), \sigma_t^2\mathbf{I})$ is adopted to approximate the ground truth reverse transition $q_{\mathbf{x}_{t-1}|\mathbf{x}_t}$. Specifically, the optimal mean value of $\mathbf{x}_t$ can be written as:

$$\mu_t^*(\mathbf{x}_t) = \frac{1}{\sqrt{\alpha_t}}\left(\mathbf{x}_t - \frac{\beta_t}{\sqrt{1-\overline{\alpha}}}\mathbb{E}[\epsilon|\mathbf{x}_t]\right) \tag{2}$$

where $\overline{\alpha_t} = \prod_{i=1}^{t}\alpha_i$, and $\epsilon$ is the standard Gaussian noises injected to $\mathbf{x}_t$. Thus, the learning is equivalent to a noise prediction task. Formally, a noise prediction network $\epsilon_\theta(\mathbf{x}_t, t)$ is used to learn $\mathbb{E}[\epsilon|\mathbf{x}_t]$ by minimizing the noise prediction objective. For $l_2$ loss, we can formulate the objective of noise prediction task as $min_\theta \mathbb{E}_{t,\mathbf{x}_0,\epsilon}\|\epsilon - \epsilon_\theta(\mathbf{x}_t, t)\|_2^2$, where $t$ is uniformly sampled between 1 and $T$. On the basis of the plain diffusion models, LDM (Rombach et al., 2022) proposes to add noise and denoise in the latent space, which greatly improves the training efficiency. Followed by LDM, U-ViT (Bao et al., 2023) proposes to replace CNN-based U-net (Ronneberger et al., 2015) with ViTs (Dosovitskiy et al., 2021) to estimate the backward process in diffusion models.

### 3.2 THE PROPOSED MoE-MLP

We propose MoE-MLP, which is a simple block composed of full MLP modules without the dot-product attention mechanism. Given an image $\mathbf{x} \in \mathbb{R}^{H_r \times W_r \times 3}$, we first use the pretrained VAE encoder to extract the latent embeddings $\mathbf{z} \in \mathbb{R}^{H_l \times W_l \times C}$ of $\mathbf{x}$, where $H_l < H_r$ and $W_l < W_r$. Then, the latent representations are divided into several patches and fed into the linear layer to obtain the input sequence $\mathbf{z} \in \mathbb{R}^{L \times D}$, where $L$ is the number of patches and $D$ is the model's hidden dimension.

**MoE-MLP blocks.** Given a sequence of tokens $\mathbf{z} \in \mathbb{R}^{L \times D}$, the input tensors are separately fed into two branches, which we call MoE-Linear Attention $MoE - FC_S$ (spatial) and $FC_C$ (channel). For the channel branch, we directly feed the embeddings $\mathbf{z}$ into $FC_C$, which can be written as:

$$\mathbf{z}_R = FC_C(\mathbf{z}) = \mathbf{z}\mathbf{W}_C + \mathbf{B}_C \tag{3}$$

where the weight $\mathbf{W}_C \in \mathbb{R}^{D \times D}$ and bias $\mathbf{B}_C \in \mathbb{R}^D$ are two learnable parameters in a linear layer.

**MoE-Linear Attention.** Different from the channel transformation, the spatial transformation plays the role of message passing between all tokens. It's difficult to learn a global token-wise interaction pattern, which is proper for all images. Therefore, we propose MoE-Linear Attention on the spatial dimension. Specifically, given the input sequence $\mathbf{z} \in \mathbb{R}^{L \times D}$, we first permute the sequence to obtain $\mathbf{z}' \in \mathbb{R}^{D \times L}$. Then, $\mathbf{z}'$ is fed into the spatial branch $MoE - FC_S$, which can be written as:

$$\mathbf{z}_L = MoE - FC_S(\mathbf{z}') = \sum Softmax(\mathbf{z}'\mathbf{W}_\sigma)(\mathbf{z}'\mathbf{W}_S) \tag{4}$$

where $\mathbf{W}_S \in \mathbb{R}^{L \times L \times E}$ and $\mathbf{W}_\sigma \in \mathbb{R}^{L \times E}$ are the learnable parameters. Both summation and $Softmax$ operation are conducted on the expert dimension. $E$ is the number of experts.

**Multi-Head MoE-MLP.** The multi-head mechanism has been proven effective on transformer-like architectures (Vaswani et al., 2017), since different heads can attend to different tokens. Inspired by

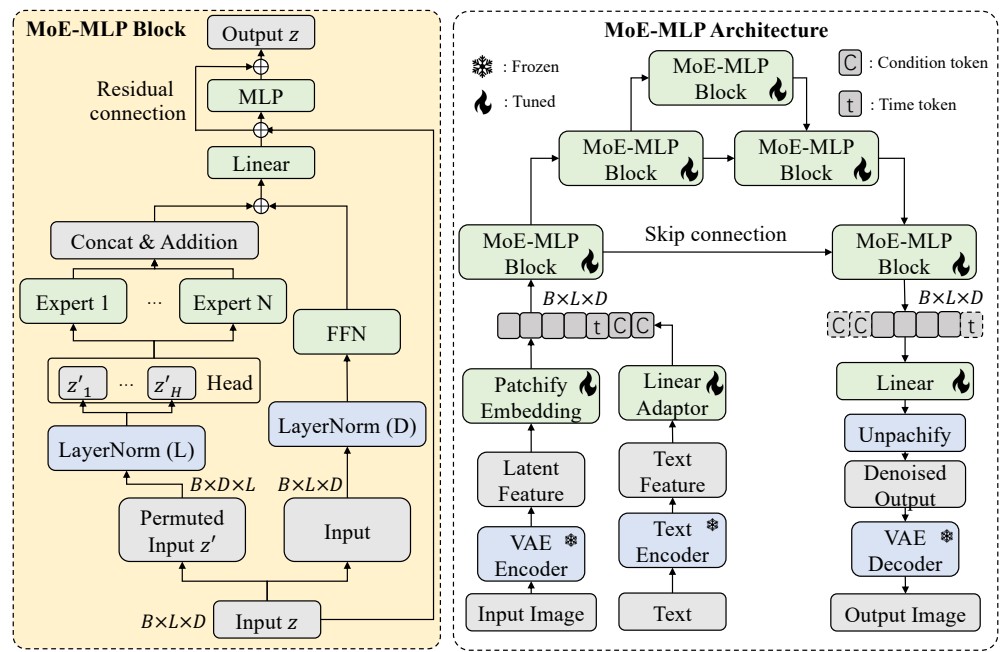

Figure 2: Framework of the proposed MoE-MLP. We illustrate the architecture of each MoE-MLP block (left) and the overall architecture of the proposed MoE-MLP (right). For simplicity, we only illustrate MoE-MLP with 5 blocks. Text and time tokens are directly concatenated with image tokens.

this, we propose multi-head MoE-Linear Attention, which divides the channel into several heads, and the features in different heads use individual experts. Specifically, given the permuted feature $\mathbf{z}'$, we first divide the feature into several heads, and we denote $\mathbf{z}'_i \in \mathbb{R}^{\frac{D}{H} \times L}$ as the features of the $i$-th head. Then, we can separately feed the features into the proposed MoE-Linear Attention, which can be written as:

$$\mathbf{z}'_i = MoE - FC_S(\mathbf{z}'_i) = \sum Softmax(\mathbf{z}'_i \mathbf{W}_\sigma)(\mathbf{z}'_i \mathbf{W}_S)$$
$$\mathbf{z}'_L = ConCat(\mathbf{z}'_1, \mathbf{z}'_2, \cdots, \mathbf{z}'_H) \tag{5}$$

where the $ConCat$ operation is conducted on the channel dimension, and $\mathbf{z}'_L \in \mathbb{R}^{D \times L}$.

**Branch Embeddings Fusion.** Similar to previous MLP-like architectures (Hu & Rostami, 2024), we directly add the two branch outcomes, followed by a linear layer, which can be expressed as:

$$\mathbf{z}_o = Linear(Permute(\mathbf{z}'_L) + \mathbf{z}_R) \tag{6}$$

where $Permute(\cdot)$ is the permutation operation to transpose the $\mathbf{z}'_L$ to obtain matrix in $\mathbb{R}^{L \times D}$. Similar to the standard transformer blocks, we use the residual connections in each block:

$$\mathbf{z} = FFN_C(\mathbf{z} + \mathbf{z}_o) + \mathbf{z} + \mathbf{z}_o \tag{7}$$

where $FFN_C$ is the Feed-forward-Neural-Networks on the channel dimension.

### 3.3 Skip Connection and Objective

**Skip connection.** Inspired by the success of the U-Net in CNN and Transformer models, our MoE-MLP also adopts similar long skip connections between the shallow and deep blocks. Intuitively, the long skip connections provide shortcuts for the low-level features and therefore ease the training of the noise prediction network. After obtaining the output of the final blocks, a $3 \times 3$ convolutional layer is employed before output. This is intended to prevent the potential artifacts in images produced by FFN (This operation is widely used in Transformer (Bao et al., 2023) and MLP-based (Hu & Rostami, 2024) diffusion models).

**Diffusion objective.** Given the latent representations $\mathbf{z}$ processed by VAE encoder, we first sample a noise $\epsilon$ and the timestep $t$ from the standard Gaussian distribution and the uniform distribution from

```
# H: num_heads, E: num_experts, D: dimension, L: sequence_length
def __init__(self, D, L, H, E, s):
    self.norm1 = norm_layer(D)
    self.norm2 = norm_layer(L)
    weight = torch.zeros(H, E, L, L)
    self.l_weight = nn.Parameter(weight)
    self.l_gate = nn.Linear(L, E)
    self.r = nn.Linear(L, E)              # For channel transformation
    self.proj = nn.Linear(D, D)
    self.mlp = Mlp(in_feat=D, hidden_feat=s*D)
# x: input tensor: B, L, D
def forward(self, x):
    xperm = x.permute(0, 2, 1)                              # B D L
    x, xperm = self.norm1(x), self.norm2(x)
    gate = self.l_gate(xperm).view(B, H, D, E)
    gate = gate.mean(dim=2).softmax(dim=-1)
    ls = einsum('bemn, bhe->bhmn', self.l_weight, gate)
    xperm = xperm.view(B, H, D, L)
    l = einsum('bhmn, bhdm->bhdn', ls, xperm)
    l = l.view(B, H * D, L).permute(0, 2, 1)
    r = self.r(x)
    x = x + self.proj(l + r)                        # Merge two branches
    x = x + self.mlp(self.norm1(x))                        # Residual
    return x
```

**Algorithm 1:** PyTorch-like pseudo code for the proposed MoE-MLP. For simplicity, $B, L, D, E$ are directly set as global variables. $s$ is the expansion factor in the MLP (channel) architecture.

$1 \sim T$, respectively. Then, we add the noise to the latent representations by:

$$\mathbf{z}_t = \sqrt{\alpha_t}\mathbf{z}_0 + \sqrt{1 - \alpha_t}\epsilon, \quad \epsilon \in \mathcal{N}(0, \mathbf{I}) \tag{8}$$

where $\alpha_t$ is the pre-defined diffusion hyperparameters. Then, the noisy input $\mathbf{z}_t$ will be patchified and fed into the proposed MoE-MLP. Finally, the diffusion objective can be written as:

$$\mathcal{L}_{Diff} = \mathbb{E}_{t, \mathbf{x}_0, \epsilon}\|\epsilon - f_\theta(\mathbf{z}_t, t)\|_2^2 \tag{9}$$

where $f_\theta(\mathbf{z}_t, t)$ is the predicted noised, and $\theta$ is the training parameters of MoE-MLP. Figure 2 shows the framework of each block and the overall architecture of the proposed MoE-MLP, and Algorithm 1 shows the PyTorch-like pseudo code of the proposed MoE-MLP block.

### 3.4 EMPIRICAL ANALYSIS

**Advantages of MoE-Linear Attention over Single Linear Attention.** We analyze why the proposed MoE-MLP architecture theoretically surpasses traditional MLP-based approaches in image generation. A key limitation of prior MLP methods is their reliance on a single, fixed matrix $W$ to approximate the attention mechanism. However, the ideal attention matrix should dynamically adapt to the diverse content distributions across different images. Forcing all images to share the same attention pattern, which inevitably leads to suboptimal token-wise information propagation, as the rigid weighting fails to capture the variability in spatial relationships (see Figure 1).

In contrast, MoE-MLP overcomes this constraint by leveraging a Mixture of Experts (MoE) to enable infinitely many attention matrix combinations. This design dramatically expands the attention mechanism's degrees of freedom, allowing the model to adaptively learn input-specific attention patterns. Consequently, MoE-MLP achieves more precise and expressive information propagation.

**Analysis on computational costs.** In summary, we introduce two novel modules for full-MLP architectures: **MoE-Linear Attention** (Spatial dimension) and the **Multi-Head** mechanism (Channel dimension). Below, we analyze the computational overhead of the proposed two modules.

- **Multi-head Mechanism.** Given an input matrix of size $D \times L$, we split it into $H$ heads, reshaping it into $H \times \frac{D}{H} \times L$. Each sub-matrix of size $\frac{D}{H} \times L$ is then multiplied by a learnable $L \times L$ matrix $\mathbf{W}$, with a computational complexity of $L^2\frac{D}{H}$ per head. Aggregating

Table 1: Efficiency comparisons between Transformer block, full MLP block, and the proposed MoE-MLP. Note that $H$ and $E$ are significantly lower than $L$ and $D$. $s$ is the expansion ratio in MLP.

| Model | Architecture | Complexity | # Params (P) | GFLOPs ↓ |
|---|---|---|---|---|
| Transformer Block (Bao et al., 2023) | ViT-S/2 | $(3 + 2s)LD^2 + 2L^2D$ | $(4 + 2s)D^2$ | 33.67 |
| MLP Block (Hu & Rostami, 2024) | ViT-S/2 | $(2 + 2s)LD^2 + L^2D$ | $(2 + 2s)D^2 + L^2$ | 17.54 |
| MoE-MLP Block (Ours) | ViT-S/2 | $(2 + 2s)LD^2 + L^2D + 2HELD$ | $(2 + 2s)D^2 + HEL^2$ | **17.55** |

Table 2: FID scores comparisons between different generative models trained on MS-COCO dataset. For MLP-based models, CNN is only used for pre- and post-processing in VAE pretrained by SD (Rombach et al., 2022). VAE is frozen in the training process. $E$ and $H$ mean the number of experts and heads, respectively.

| Model | FID ↓ | Type | Model Architecture | #Param. |
|---|---|---|---|---|
| AttnGAN (Xu et al., 2018) | 35.49 | GAN | CNN + Attention | 230M |
| DM-GAN (Zhu et al., 2019) | 32.64 | GAN | CNN + Memory Network | 46M |
| VQ-Diffusion (Gu et al., 2022) | 19.75 | Diffusion | CNN + Transformer | 370M |
| XMC-GAN (Zhang et al., 2021) | 9.33 | GAN | CNN + Attention | 166M |
| LAFITE (Zhou et al., 2022) | 8.12 | GAN | CNN + Transformer | 75M + 151M |
| LDM (Rombach et al., 2022) | 7.32 | Latent diffusion | CNN + Cross-attention | 53M (Backbone) + 207M (VAE) |
| DiT-S/2 (Peebles & Xie, 2023) | 6.23 | Latent diffusion | CNN + Transformer | 45M (Backbone) + 207M (VAE) |
| U-ViT-S/2 (Bao et al., 2023) | 5.95 | Latent diffusion | CNN + Transformer | 45M (Backbone) + 207M (VAE) |
| **MLP-based model** | | | | |
| gMLP (Liu et al., 2021) | >100 | Latent diffusion | CNN + MLP | 45M (Backbone) + 207M (VAE) |
| MLP-Mixer (Tolstikhin et al., 2021) | >100 | Latent diffusion | CNN + MLP | 45M (Backbone) + 207M (VAE) |
| UL-MLP (Hu & Rostami, 2024) | 8.62 | Latent diffusion | CNN + MLP | 47M (Backbone) + 207M (VAE) |
| MoE-MLP ($2E1H$) (Ours) | 7.61 | Latent diffusion | CNN + MLP | 49M (Backbone)s + 207M (VAE) |
| MoE-MLP ($4E1H$) (Ours) | 7.51 | Latent diffusion | CNN + MLP | 52M (Backbone)s + 207M (VAE) |
| MoE-MLP ($4E2H$) (Ours) | **7.43** | Latent diffusion | CNN + MLP | 60M (Backbone)s + 207M (VAE) |

across all $H$ heads, the total complexity remains $L^2D$-identical to non-head (single-matrix) version. Thus, the multi-head design **incurs no additional computational cost**.

- **MoE-Linear Attention.** For the $D \times L$ input matrix $\mathbf{z}$, we first route it through gating linear layers with weight matrices of size $L \times H \times E$, where $E$ denotes the number of experts and $H$ is the number of heads (Note that different heads do not share the experts). The input is then processed by these $E$ expert models. Fortunately, for linear combination, we have:

$$\sum_i \lambda_i \mathbf{w}_i \mathbf{z} = (\sum_i \lambda_i \mathbf{w}_i)\mathbf{z} = \mathbf{W}_{MoE-Linear}\mathbf{z} \qquad (10)$$

where $\lambda$ is the gating matrix. Then, we can first derive the MoE-Linear attention matrix $\mathbf{W}_{MoE-Linear}$, and these calculations result in $HELD$ computational costs. Finally, the attention matrix multiplies the input matrix, which results in the total complexity of $L^2D$. Therefore, our method only brings an extra $HELD$ computational complexity.

We report the complexity of transformer block (Bao et al., 2023), traditional MLP block (Hu & Rostami, 2024) and the proposed MoE-MLP ($E = 4$, $H = 1$) in Table 1. Although our method introduces more parameters ($(HE - 1)L^2$) for each block than traditional MLP blocks, the overall computation increases by only 0.01 GFLOPs. Furthermore, the computational cost of each Transformer block is nearly double that of a single block in our proposed MoE-MLP. Notably, as the attention mechanism has matured, there are now several open-source libraries (*e.g.*, xFormers (Lefaudeux et al., 2022), Flash Attention (Dao et al., 2022; Dao, 2024)) designed to accelerate attention computation, which leads to faster training speeds (iterations per second) compared to MLPs.

## 4 EXPERIMENTS

### 4.1 MAIN RESULTS

**Main results.** We report the FID results of our method under different numbers of experts and heads in Table 2. The experimental results show that, with the same number of parameters, our method achieves comparable performance to the CNN-based architecture LDM (Rombach et al., 2022). Moreover, by adding only a single expert (*i.e.*, introducing just roughly one additional

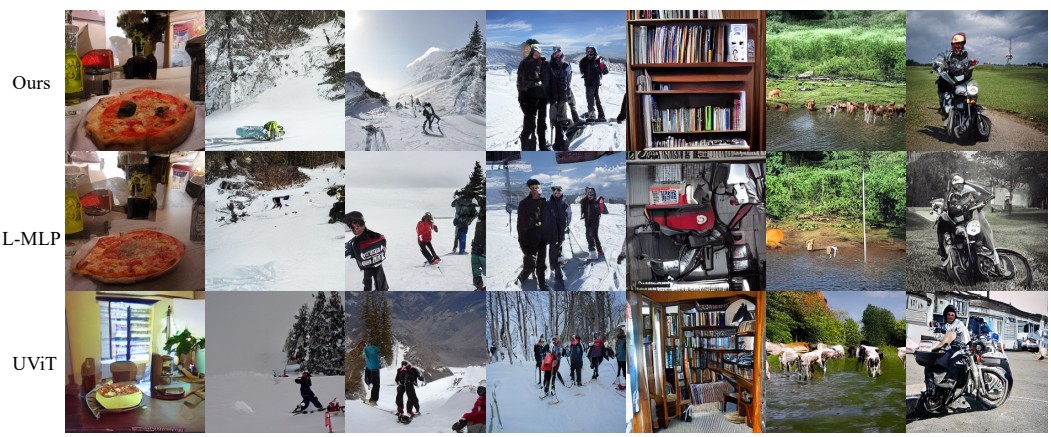

Figure 3: Qualitative comparisons between the proposed MoE-MLP (4 Experts and 1 Head), U-ViT (Bao et al., 2023) and L-MLP (Hu & Rostami, 2024). All methods are trained on the MS-COCO.

Table 3: FID scores comparisons between different model architectures trained on MS-COCO dataset.

| Model | FID ↓ | Complexity | # Params | #Speed (Iter. / Sec.) |
|---|---|---|---|---|
| Arch. (Fig. 8a) | 124.282 | $2HELD + 2sLD^2 + L^2D$ | 48M (Model) + 207M (VAE) | 17.371 |
| Arch. (Fig. 8b) | 16.482 | $2HELD + (4s+2)LD^2 + L^2D$ | 62M (Model) + 207M (VAE) | 14.918 |
| Arch. (Fig. 8c) (Ours) | **14.973** | $2HELD + (2s+2)LD^2 + L^2D$ | 52M (Model) + 207M (VAE) | 17.125 |

learnable matrix with shape $L \times L$ per block), we achieve an FID improvement of **1.01** over the original MLP-based method (Hu & Rostami, 2024). Figure 3 shows the qualitative comparisons of the proposed MoE-MLP with previous MLP-based method (Hu & Rostami, 2024) and U-ViT (Bao et al., 2023), where the images generated by MoE-MLP are of similar quality to those generated by U-ViT, while the images generated by L-MLP tend to have more noise artifacts. We also illustrate more images generated by our MoE-MLP in Appendix A for further comparisons.

## 4.2 ABLATION STUDIES

**Effect on heads. Setups.** To systematically evaluate the impact of multi-head attention mechanisms in our model, we conducted a comprehensive ablation study by varying the number of attention heads while keeping the number of experts fixed. Specifically, we compared configurations with head counts ranging from 1 to 8 to assess their influence on model performance. Due to computational constraints, we limited the training to 100,000 iterations with a fixed learning rate of 2e-5. The training process utilized a batch size of 256, distributed across 8 GPUs (32 samples per GPU) to optimize parallel efficiency. During inference, we adopted a Classifier-Free Guidance (CFG) ratio of 1.0 for stable sampling. To ensure robust evaluation, we randomly sampled 30,000 generated images and computed their Fréchet Inception Distance (FID) against the validation set, providing a reliable measure of image quality and diversity. **Results.** As illustrated in Figure 4, our experimental results demonstrate that increasing the number of attention heads leads to consistent and significant improvements in FID compared to the baseline model (1 Expert, 1 Head). Notably, when using a smaller number of heads (e.g., 2–4), our approach achieves competitive performance gains without introducing substantial additional parameters or significantly increasing computational overhead. This suggests that our method maintains an efficient trade-off between model capacity and computational cost, particularly in scenarios where resource efficiency is critical. These findings highlight the effectiveness of our architecture in leveraging multi-head attention while preserving scalability, making it suitable for practical applications where both performance and efficiency are key considerations.

**Effect on experts. Setups.** We conducted a systematic investigation into the impact of varying the number of experts in our proposed MoE-Linear framework. To isolate the effect of expert count, we held the number of attention heads constant (fixing it at either 1 or 2) while progressively increasing the number of experts from 1 to 8. Consistent with our previous experiments, all models were trained for 100,000 iterations on the MS-COCO dataset under identical hyperparameter settings.

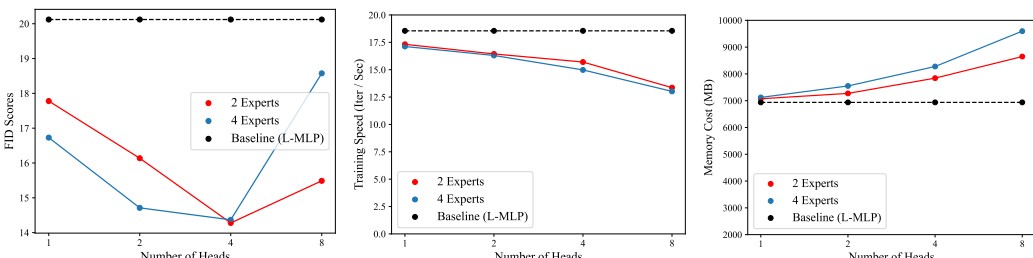

Figure 4: FID (left plot), training speed (middle plot), and memory costs (right plot) comparisons of different numbers of heads with baselines (Hu & Rostami, 2024). We fix the number of experts and change the number of heads (1 ∼ 8). For FID comparisons on the MS-COCO dataset, we randomly sample 30K images.

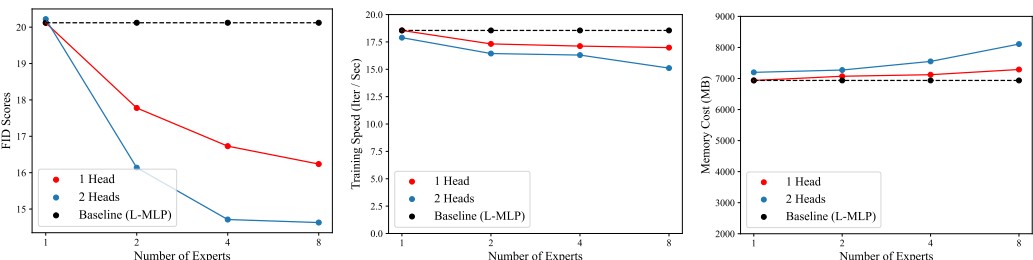

Figure 5: FID (left plot), training speed (middle plot), and memory costs (right plot) comparisons of different numbers of experts with L-MLP. We fix the number of experts and change the number of heads (1 ∼ 8). For FID comparisons on the MS-COCO dataset, we randomly sample 30K images.

**Results.** As demonstrated in Figure 5, we observed several key findings: 1) The FID score exhibited consistent improvement as we increased the number of experts, indicating enhanced model capacity for image generation; 2) This performance gain came with a gradual increase in both training time and parameter count, attributable to the linear growth of $L \times L$ transformation matrices with respect to the number of experts. Notably, even when scaling to 8 experts, our MoE-MLP architecture maintained remarkable parameter efficiency. The most significant performance leap occurred when increasing from 1 to 2 experts, where the FID score improved dramatically from 20.121 to 17.779 at 100k iterations - a substantial gain achieved by adding just one additional $L \times L$ transformation matrix for each MoE-MLP block. These findings collectively validate that our MoE-Linear modification represents an efficient and effective method for enhancing traditional full MLP-based architectures, achieving significant quality improvements with minimal computational overhead.

**Model Architecture. Setups.** To further validate our design choices, we conducted a comprehensive ablation study examining different network architectures for latent diffusion models. As illustrated in Figure 8 in the Appendix A, we evaluated three distinct configurations: (a) Modified Transformer Baseline: We replaced the standard attention mechanism in a conventional transformer architecture with our proposed MoE-Linear Attention, incorporating an additional softmax operation on the parameters. This variant served to investigate the compatibility of our approach with traditional transformer frameworks. (b) Parallel MLP Branch Design: This architecture introduced two parallel MLP branches, with their outputs concatenated and fused through a linear projection layer. The design aimed to explore alternative feature integration strategies while maintaining the MoE-Linear components. (c) Proposed Architecture: Representing our main experimental configuration (as detailed in Section 3), this architecture served as the reference for comparative evaluation. For all variants, we maintained consistent hyperparameters: 4 experts and 1 head in each MoE-Linear Attention layer, trained for 100k iterations on the MSCOCO dataset. The quantitative comparisons are presented in Table 3. **Results.** Variant (c) achieved superior results, demonstrating the effectiveness of our proposed architecture. The significant performance gap (FID difference of $\Delta = 3.21$) between variants (c) and (a) suggests that simply transplanting MoE-Linear Attention into traditional transformer frameworks may be suboptimal. The particularly poor performance of variant (a) indicates potential incompatibility between softmax normalization and fully MLP-based architectures. This

observation aligns with recent findings in L-MLP (Hu & Rostami, 2024) regarding attention-like operations in full MLP models.

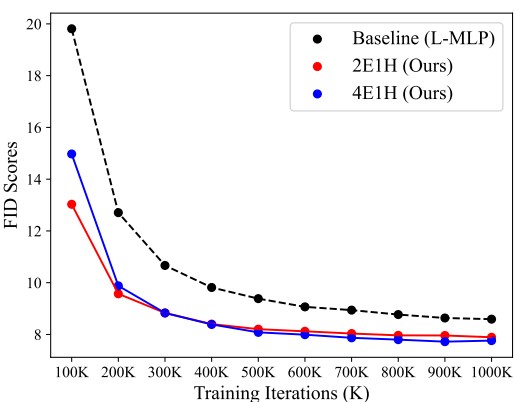

Figure 6: FID scores of L-MLP and the proposed MoE-MLP with different training iterations.

**Convergence speed.** To better compare the convergence speed of our proposed MoE-MLP with the baseline L-MLP, we report the FID results of both models at different training iterations. As shown in Figure 6, our method achieves an FID score of 13.029 at 100K training iterations, while the baseline L-MLP only reaches an FID score of 19.810. This improvement can be attributed to the MoE Linear Attention architecture we proposed, which allows the attention matrix to have an infinite number of possible combinations. In contrast, the baseline L-MLP (Hu & Rostami, 2024) only provides the same attention matrix for all images. Our approach significantly increases the degrees of freedom for the attention matrix, leading to better results. Additionally, we observed that although the model with 2 Experts achieves better results than the one with 4 Experts after 100 training iterations, the performance of the model with 4 Experts surpasses that of the 2 Experts model as training progresses. This can be attributed to the fact that the 2 Experts settings may exhibit faster convergence, whereas the 4 Experts setup provides greater flexibility, allowing the network to automatically learn a broader range of possible Linear Attention combinations in later stages of training. As a result, the attention matrix becomes more precise, leading to improved overall performance on the MS-COCO dataset.

**Using load balancing loss. Setups.** The load balancing objective consists of two key components: **i) Coefficient of Variation (CV) term**, which penalizes imbalanced expert usage by measuring the normalized standard deviation of gating probabilities, and **ii) Cross-entropy term**, which can be written as: $\mathcal{L}_{Balance} =$

Table 4: FID scores comparisons on the MS-COCO dataset w/ and w/o load balancing loss.

| Method | Training Speed (Iter. / Sec.) | # Iterations | FID ↓ |
|---|---|---|---|
| w/ Balance | 13.85 | 100K | **11.757** |
| | | 1M | 8.557 |
| w/o Balance | 14.27 | 100K | 12.205 |
| | | 1M | **8.018** |

$\left(\frac{\sigma_u}{\mu_u+\alpha}\right)^2 - \sum_{e=1}^{E} \frac{1}{E} \log(u_e + \alpha)$, where $u_e$ is the mean gating probability for $e$-th expert, $\mu_u = \frac{1}{E} \sum_{e=1}^{E} u_e$ is the mean expert usage. $\alpha$ is a small constant for numerical stability. Although the load balancing loss works well for traditional large language models, in our scenario, MoE-MLP aims to provide more combinations of the attention matrix, and the load balancing loss may cause different combination weights to be equal. Therefore, in our main experiments, we discard this balancing term. For comparisons, we also report the FID scores w/ and w/o load balancing loss. Following the work (Shazeer et al., 2017), we set the balancing loss weight as 0.01, training the MoE-MLP with different iterations. Table 4 shows comparisons of FID results at 100k and 1M training iterations. We observe that with load balancing loss, the model converges faster (achieving better FID at 100k), whereas at 1M training iterations, "w/o load balancing loss" performs better.

## 5 CONCLUSION

In this paper, we propose MoE-MLP, which includes two main modules: i) MoE-Linear Attention, which is composed of several learnable attention matrices, and adaptively assigns learnable weights to each matrix to every image; ii) Multi-Head module, which partitions the original channels into multiple heads and performs MoE-Linear Attention on each head separately. The two modules significantly increase the diversity of the final attention matrix. Then, we leverage the property of linear combination, reducing the complexity of the MoE-Linear Attention to $L^2D + HELD$, which only brings negligible additional computations. Finally, we conduct experiments on the text-to-image generation task, where the results demonstrate the effectiveness of the proposed MoE-MLP.

**LLM-Usage Statement.** The authors used a large language model for language polishing. All ideas, methodology, experiments, and results are the authors' own.

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

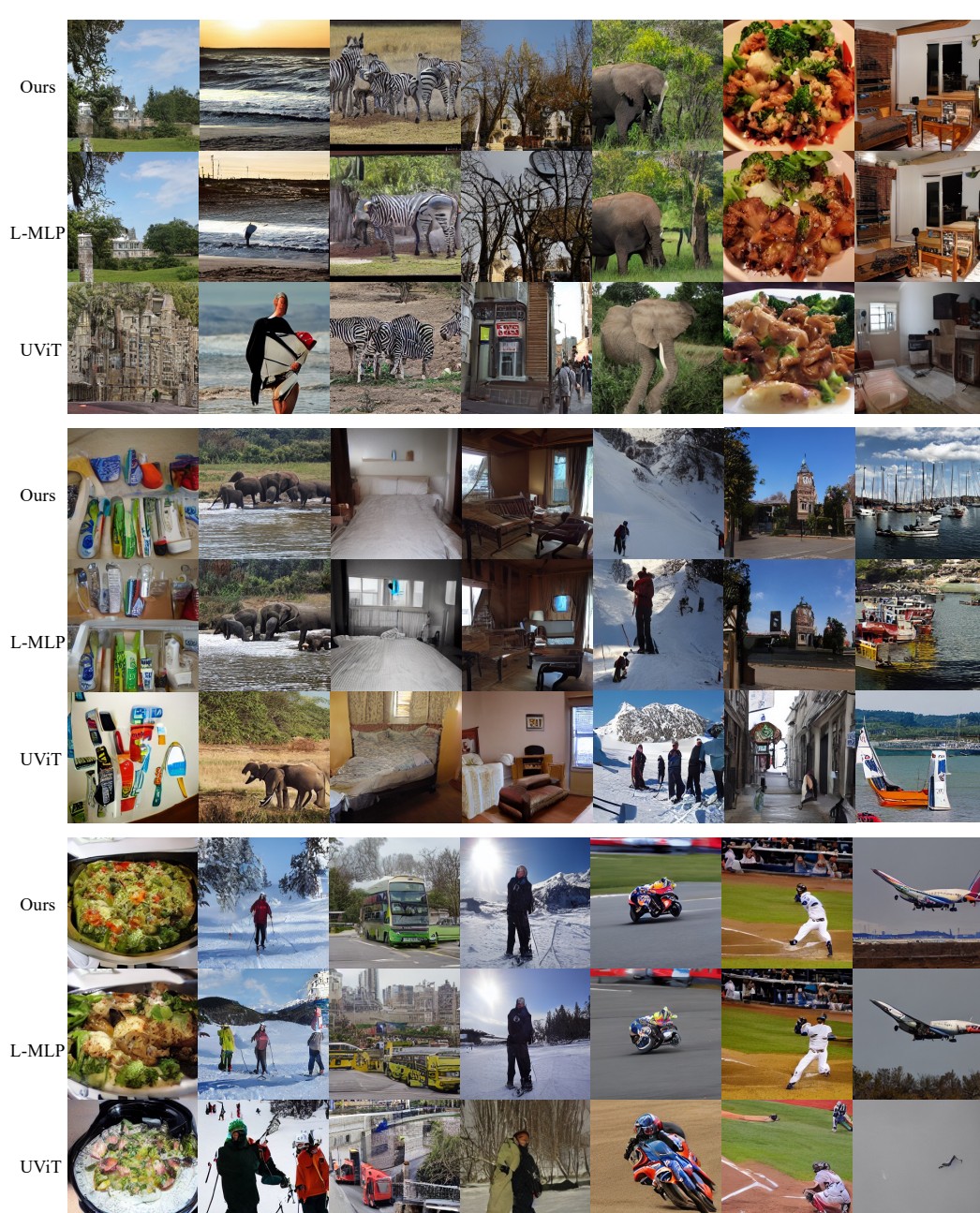

Figure 7: More qualitative comparisons between the proposed MoE-MLP, U-ViT, and L-MLP.

## A EXPERIMENTAL SETUPS

To evaluate the effectiveness of the proposed MoE-MLP, we conduct experiments on text-to-image synthesis task using the MS-COCO dataset Lin et al. (2014). The dataset consists of $256 \times 256$ resolution images, split into 82,783 training samples and 40,504 validation samples. Each image is paired with five captions, and during training, similar to the work Bao et al. (2023), a random caption is selected for each image. We generate images from 30,000 randomly chosen captions in the validation set, and we mainly compare our MoE-MLP with transformer-based methods Bao et al. (2023) and traditional MLP-based methods Hu & Rostami (2024).

Our MoE-MLP is built upon the UL-MLP Hu & Rostami (2024) diffusion framework, where timestep embeddings, text conditions, and image features are uniformly processed as tokenized inputs. Following U-ViT's preprocessing pipeline, we encode images into 512-dimensional latent features using a pretrained autoencoder, followed by a $2 \times 2$ convolutional upscaling layer. Text inputs are embedded into a $77 \times 512$ representation using a pretrained CLIP model, with an additional linear projection layer for adaptation. During training, we apply classifier-free guidance by randomly replacing text embeddings with null tokens 10% of the time. The model is trained on 8 GPUs (H800) with a batch size of 256.

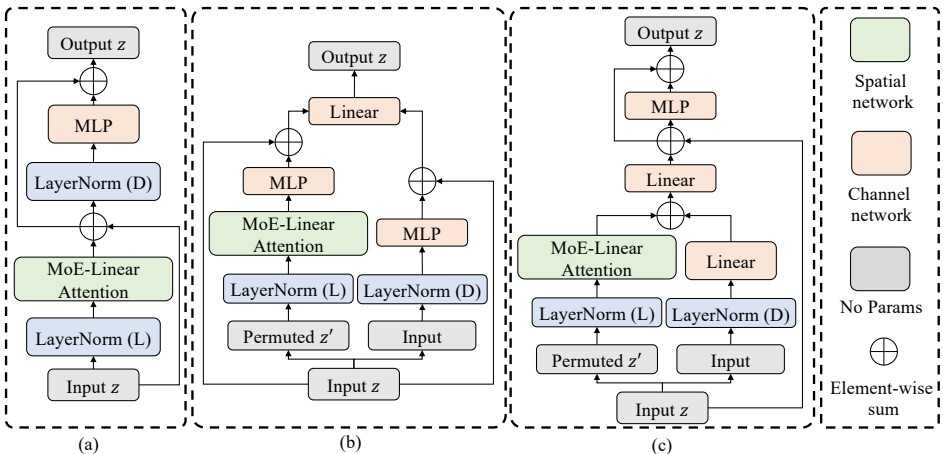

Figure 8: Visualization of three different designs of the model architectures.

## B   MORE QUALITATIVE COMPARISONS

As shown in Figure 7, since both our MoE-MLP and L-MLP Hu & Rostami (2024) are MLP-based methods, the images generated by these two approaches tend to have similar content. However, the images generated by our proposed MoE-MLP exhibit higher-quality details and fewer noise artifacts.

