# OpenReview forum: "Many Can Beat One: MoE-Linear Attention for Full MLP Image Generation"
_ICLR.cc/2026/Conference — ICLR 2026 Conference Withdrawn Submission_

### Official Review · Reviewer_7F8e · 2025-10-16

**Soundness:** 2
**Presentation:** 2
**Contribution:** 2
**Rating:** 4
**Confidence:** 3

**Summary:**

The paper proposes a novel architecture, MoE-MLP, which enhances traditional full-MLP models for image generation by introducing a Mixture of Experts (MoE) mechanism to create dynamic, input-adaptive "attention" matrices. The core idea is compelling: instead of a single, global token interaction matrix, the model learns a small set of expert matrices and linearly combines them with input-dependent weights. This is complemented by a multi-head mechanism to increase diversity. The method improves performance (FID of 7.43 vs. baseline 8.62) with negligible computational overhead on MS-COCO dataset for MLP-based latent diffusion methods, but is still behind transformer-based latent diffusion models with a similar model scale.

Given this approach is only better than the MLP baseline but likely still behind transformer-based methods that was introduced several years ago even if scaled to the same computational cost, I suggest to reject.

**Strengths:**

1: The method improves the performance of MLP-based latent diffusion models, achieving lower FID score on MS-COCO dataset.

2: The method does not introduce a lot of new computational cost  compared with MLP Block despite it introduces more parameters.

3: The authors perform a lot of detailed ablation studies.

**Weaknesses:**

1: The performance of the proposed method is still far from transformer-based methods with similar parameters which was published 2 years ago. Although the proposed method costs less computation, the authors actually do not perform experiments on a fairly-compared setting which both methods cost the same FLops. According to table 2, the benefits of scaling up the proposed methods seem marginal, so I assume the method is still behind transformer-block even if scaled to same FLops.

2: The authors only evaluate the method on one metric ,FID, on one dataset, MS-COCO. Usually it is necessary to evaluate on more metrics(for example, human preference) and generalizing the method on more datasets.

3: The naming and descriptions of the methods are hard to understand. The authors do not produce a normal latex algorithm, but instead offers a pusedo code without detailed comments. Also the naming of the proposed method is very confusing. MoE seems from "Mix of Expert" in transformers but the proposed method is different from MoE transformers, which selects a few experts instead of concating all experts as shown in figure 2. Also equation (5) says "concat" but figure 2 illustrates "concat and addition", causing contradiction.

**Questions:**

1: Which one correctly describes the model, equation 5 or figure 2?

2: What's the performance of the proposed method if we scale it to the same FLops of Bao et. al, 2023?

---

### Official Review · Reviewer_Kt1X · 2025-10-27

**Soundness:** 3
**Presentation:** 3
**Contribution:** 3
**Rating:** 4
**Confidence:** 4

**Summary:**

This paper proposes a MoE-Linear Attention module and a Multi-Head module to enhance the token interaction capabilities of full MLP architectures in image generation tasks.. The method is evaluated on MS-COCO text-to-image generation and shows clear improvements over existing MLP-based methods, achieving competitive FID scores with minimal computational overhead.

**Strengths:**

1.The writing structure of the paper is relatively well-organized.

2.The experiments in the paper are relatively comprehensive.

3.The proposed technique shows relatively strong effectiveness compared to the baseline methods.

**Weaknesses:**

1.The compared methods in the paper are relatively outdated. Are there any more recent MLP-based approaches that could be included for comparison?

2.The description of the method is somewhat unclear, especially in the section introducing the MoE component.

3.The final FID and image quality achieved by the proposed method appear to be quite far from the current state-of-the-art image generation approaches.

**Questions:**

1.Could you elaborate on the differences between the MoE component and the Multi-Head mechanism? It seems that the MoE here is fully activated — is that the case?

2.Is there more detailed analysis or visual evidence demonstrating that the use of MoE improves adaptability for generating images from different categories or domains, rather than simply increasing the number of parameters?

3.Could you include comparisons with more recent MLP-based image generation methods?

4.It appears that current MLP-based methods do not show significant advantages over state-of-the-art image generation techniques. Could you briefly comment on this point?

Note: My main concerns are points 1 and 2 — if these can be addressed effectively, I can raise the score.

---

### Official Review · Reviewer_5AFx · 2025-10-30

**Soundness:** 3
**Presentation:** 2
**Contribution:** 2
**Rating:** 4
**Confidence:** 4

**Summary:**

To reduce the substantial computation overhead of Transformer-based models and enhance the performance of MLP-based models in image generation tasks, the paper proposes MoE-MLP, including two modules: 1) MoE-Linear Attention, which is composed of several learnable attention matrices, and adaptively assigns learnable weights to each matrix to every image; 2) Multi-Head module, which partitions the original channels into multiple heads and performs MoE-Linear Attention on each head separately. Experimental results show that the proposed method achieves an FID of 7.43 on MS-COCO, outperforming traditional MLP-based models by 1.19.

**Strengths:**

1.The MoE and Multi-Head mechanisms are innovatively introduced into MLP-based architectures, enhancing the diversity of token interactions within MLP-based models. The methodology is clearly illustrated in the paper, with a well-structured presentation supported by formulas, diagrams, and pseudocode.
2.The paper conducts extensive comparative and ablation experiments, thoroughly demonstrating the effectiveness of the proposed MoE-MLP architecture.

**Weaknesses:**

Although the authors have done an excellent job, some issues in the paper are worth exploring:
1.Some important experimental results in the paper are not sufficiently explained:  (1) When analyzing the effects on heads, why does performance improve with 1–4 heads but degrade when increasing from 4 to 8 heads?  (2) In the architectural comparison, the paper only discusses the differences and performance between architectures A and C, while omitting analysis of architecture B, which is more similar to C.
2.The paper lacks experimental validation of its core motivation. The MLP-based architecture is proposed to alleviate the substantial computational overhead of Transformer-based models, however the paper only provides theoretical complexity analysis without empirical verification. For instance, it does not compare the actual inference efficiency between MoE-MLP and Transformer models under comparable settings—such as at similar parameter scales or equivalent performance levels.
3.The proposed method lacks novelty. On one hand, both MoE and the Multi-Head mechanism are well-established concepts that have been widely adopted in Transformer-based models. The authors merely adapt and integrate these existing techniques into an MLP-based framework without introducing further innovation. On the other hand, while MLP-based models may partially reduce the computational burden of Transformers, state-of-the-art Transformer-based models (e.g., DiT) demonstrate strong scalability—significantly improving performance as model size increases—and can be accelerated via distillation, quantization, and other techniques. It remains unclear whether MLP-based architectures can achieve similar scalability, raising questions about their long-term potential.

**Questions:**

See Weaknesses part.

---

### Official Review · Reviewer_wTNu · 2025-11-02

**Soundness:** 3
**Presentation:** 3
**Contribution:** 3
**Rating:** 4
**Confidence:** 3

**Summary:**

The paper proposes MoE-MLP, an attention-free backbone that replaces per-token dot-product attention with a Mixture-of-Experts Linear Attention and a multi-head design. The authors conduct experiments on MS-COCO datasets, and the results demonstrate that our method achieves 7.43 FID.

**Strengths:**

1. Clear writing: The MoE–Linear formulation and multi-head split are precisely specified (Eq. 4–7) with a concise pseudo-code block.

2. Systematic studies over heads/experts and a convergence plot show faster early progress vs L-MLP.

**Weaknesses:**

1. The claimed benefits over softmax attention are evaluated in a regime where sequence length is short. At 256×256 input, SD-VAE 8× downsampling yields a 32×32 latent; with 2×2 patching this is only L = 16×16 = 256 tokens. Many practical gains (and losses) of linear/MLP-style interactions versus softmax attention emerge when L is large—both for modeling long-range dependencies and for the compute/memory trade-off (softmax ∼ O(L²) vs linear/MLP ∼ O(L)). As a result, the current experiments do not probe the regime where these architectures meaningfully diverge. I recommend adding higher-resolution studies.

**Questions:**

Results are only on MS-COCO 256×256; no higher resolutions, additional datasets (LAION subset, Parti-Prompts, etc.), or metrics (IS/CLIP-A/Precision-Recall).

---

### Note · Authors · 2025-11-22

I have read and agree with the venue's withdrawal policy on behalf of myself and my co-authors.